# In Vitro Bioactivities of Plants Used against Skin Diseases in the Eastern Free State, South Africa

Valeria Makhosazana Xaba [1], Ariyo Lateef Adeniran [2], Siphamandla Qhubekani Njabuliso Lamula [3] and Lisa Valencia Buwa-Komoreng [1,3,*]

1 Department of Plant Sciences, University of the Free State, Private Bag X13, Phuthaditjhaba 9866, South Africa; valeriaxaba@gmail.com
2 Department of Biochemistry and Physiology, Faculty of Veterinary Medicine, University of Abuja, Abuja Airport Road FCT—Abuja P.M.B 117, Abuja 900105, Nigeria; lateef.adeniran@uniabuja.edu.ng
3 Botany Department, Faculty of Science and Agriculture, University of Fort Hare, Private Bag X1314, Alice 5700, South Africa; slamula@ufh.ac.za
* Correspondence: lbuwa@ufh.ac.za; Tel.: +27-40-602-2648

**Abstract:** Skin diseases are a worldwide issue, accounting for approximately 34% of all occupational illnesses. The aim of this study was to investigate medicinal plants used to treat wounds and skin diseases in the eastern Free State Province of South Africa. An ethnobotanical survey was conducted to gather information from traditional healers on plants they use to treat human ailments. Plants were collected and then investigated for antimicrobial, antioxidant and anti-inflammatory properties using standard assays. *Cotyledon orbiculata*, *Dioscorea sylvatica*, and *Lycopodium clavatum* had the highest frequency of citation (RFC) values among the 22 plants reported. Saponins, flavonoids, terpenoids, alkaloids, anthraquinones, and tannins were found in the phytochemical examination. *L. clavatum* had the greatest activity against *Staphylococcus aureus* and *Pseudomonas aeruginosa* at 0.39 mg/mL and 0.098 mg/mL, respectively. *C. orbiculata* and *D. sylvatica* extracts showed significant antifungal activity between 0.39 and 1.56 mg/mL. Antioxidant activity against 1,1-diphenyl-2-picrylhydrazyl radicals was found in all extracts. The extracts had significant anti-inflammatory action against the 5-lipoxygenase enzyme, with $IC_{50}$ values ranging from 0.02 to 0.49 g/mL. The usage of *C. orbiculata*, *D. sylvatica*, and *L. clavatum* in the treatment of skin problems in the Eastern Free State of South Africa was verified in this research.

**Keywords:** skin diseases; phytochemical; antimicrobial; antioxidant; 5-lipoxygenase assay





## 1. Significance of the Main Findings

Our findings revealed that the medicinal plants screened in this study possess phytochemical constituents that support pharmacological activities in plants. The observed antimicrobial, antioxidant, and anti-inflammatory activities of these plants demonstrate that they can be selected in the development of potential antimicrobial drugs that can be used in the management of skin diseases.

## 2. Background

Internally and externally, the skin covers the whole surface of the body [1]. The skin's reactions to the environment are significant contributors to both external and internal elements. Its most important purpose is to protect the body by acting as a defensive and mechanical barrier to the surrounding environment, preventing disease infiltration. It also helps keep bacteria that have an impact on human health and illness alive [2]. The skin is inhabited by diverse microbial flora, including *Staphylococcus* sp., *Propionibacterium*, *Diphtheroids*, *Micrococcus*, *Bacillus* sp., and a few fungal species [3,4]. These bacteria, known as normal flora, help keep the skin healthy by acting as competitive inhibitors of pathogenic germs [5].

Although skin serves as a protective barrier, it may still be damaged, allowing opportunistic microbial organisms to infiltrate the skin [5]. For example, if *Staphylococcus epidermidis*, a helpful commensal bacterium of the skin, penetrates the skin's integrity and enters the blood circulatory system, it may be lethal [5]. This type of breach can create an entry route for the microbial flora to generate an infection which can invade deep underlying tissues [1]. Leg ulcers, burns, and surgical or traumatic wounds are all examples of infections that enable a broad spectrum of bacteria to enter and colonize [4]. The most common early colonizers of burn wounds include *S. aureus*, *Escherichia coli*, and *Streptococci* sp. [6,7]. According to Armstrong [8] and Hansis [9], in polymicrobial wound infections, distinguishing between causal pathogenic and non-pathogenic species is difficult since the responsible microorganisms or etiologic agents cannot be simplified because of the presence of mixed pathogens, commensal or facultative and potentially virulent agents. Another group of colonizers such as *Pseudomonas aeruginosa*, along with *Enterococcus* sp. and *Bacillus* sp., cause invasive burn infections in patients with weak immune systems and injectable drug users [4,5]. Other pathogens, including those among the β-haemolytic *Streptococci* group A, *Klebsiella pneumoniae*, *Acinetobacter baumanni*, and *Candida albicans*, cause a wide variety of diseases [3,4,10,11]. Skin disorders range in severity from benign ailments that affect the look of the skin to severe conditions that cause deformity, disability, and anguish or even death [12]. Wounds and burns on the skin have recently been identified as some of the most severe illnesses and infections with high fatality rates. The great majority of skin disease cases are caused by a few prevalent disorders [13]. Physician visits, hospital treatment, prescription medications, and over-the-counter (OTC) items used to treat or manage skin disorders, as well as indirect expenses such as lost productivity, all add up to a major financial burden. Davis [14] argues that OTC bandage products are efficient and that they contain antimicrobials. They have been used for decades for quicker healing processes, such as covering and protecting wounds, but they may not be effective in managing and reducing the invasion of bacteria in wounds. Moreover, these products pose a disadvantage to the underprivileged communities in developing countries since they are inaccessible and expensive, and there is a shortage of appropriate products used in treating wounds and skincare [15]. Annually, about 19,500 fire-related fatalities are recorded in South Africa, making it one of the top 15 causes of mortality among children and young people aged 5 to 29 years [16]. Infections are responsible for 50–75% of burn-related fatalities [17]. The majority of people, particularly in rural areas, still rely heavily on medicinal herbs to cure skin ailments [16]. Many studies have been undertaken in the search for antimicrobial, anti-inflammatory, and other therapeutic agents which have the ability to reduce both internal and external swelling [15,18]. Such interests focused on medicinal plants which possess phytochemicals that are known to exhibit potent anti-inflammatory effects [19,20]. In this study, extracts prepared from plants used in traditional medicine in the eastern Free State for various skin ailments were tested for the presence of phytochemicals and antimicrobial, antioxidant, and anti-inflammatory properties.

## 3. Results

### 3.1. Ethnobotanical Survey

Eight females of ages ranging between 50 and 65 years and two males aged 35 to 55 years were randomly selected and interviewed. The local names of the plants, their applications, the portions of the plant utilized, the manner of preparation, the composition, and the method of administration were all compiled. A total of 22 plant species from 18 families were reported to be employed in the treatment of wounds and skin diseases (Table 1).

**Table 1.** List of medicinal plants used for the treatment of wounds and skin infections in eastern Free State Province, South Africa.

| Scientific Names | Family | Common Names | Plant Part Used | Medicinal Uses | Applications and Preparations |
|---|---|---|---|---|---|
| *Acorus calamus* L. | Acoraceae | Sweet flag (E), uKhalumuzi (Z), and Makkalmoes (A) | Rhizome | Used for stomach ulcers, chest complaints, asthma, headache, appetite stimulant, antibacterial, mucus congestion, and diarrhoea. | Preparations: Dried or candied rhizomes may be swallowed straight or in a boiling water infusion. Typically, alcoholic extracts are employed. Application: Taken via oral administration. |
| *Carpobrotus edulis* (L.) L.B. Bolus. | Aizoaceae | Sour fig/Cape fig (E), Umgongozi/Ikhambilama-bulano (Z), and Hottentosvyg (A) | Leaves/roots | Used for oral and mouth ulcers, burns, bruises, scrape, cuts, eczema, dermatitis, and other skin conditions. Internally, it is used to treat diarrhoea, dysentery, and stomach pains, as well as laryngitis, sore throats, and mouth infections. | Preparations: The leaf pulp may be used to heal wounds and infections on the skin. The leaf juice is astringent and antibacterial in nature. Application: The juice is orally consumed or used as a mouth wash. Chewing a leaf tip and drinking the liquid might help with a sore throat. The leaf juice is used to burns, sunburns, and other skin conditions as a lotion. |
| *Aloe ferox* Mill. | Aloaceae | Bitter aloe (E) Umhlaba (Z), and Ikhala (X) | Leaves and roots | Used for dermatis, cutaneous, disorders of skin, burns, and jaundice. Also helps with toothaches, earaches, and oral and vaginal thrush, as well as skin and wound healing. | Preparations: The leaves are cooked in boiling water, cooled, and filtered before being given orally in a half-cup dose. Skin wounds and infections such as burns, bruises, scrapes, cuts, sunburn, eczema, dermatitis, and other skin disorders may be treated with the leaf pulp. The leaves are used as an emetic and laxative. Approximately three leaves. Applications: Taken via oral administration. The root infusion's fresh juice is eaten orally or gargled. |
| *Haemanthus albiflos* Jacq. | Amaryllidaceae | Paintbrush (E) and uZaneke | Roots | Used for sores and wounds. | Preparations: Powdered sun-dried roots are infused in water and consumed orally to treat wounds. Applications: Taken via oral administration. |
| *Alepidea amatymbica* var. *amatymbica* Eckl. & Zeyh. | Apiaceae | Larger tinsel (E), iKhathazo (Z), and Iqwili (X). | Roots/rhizomes | Used for inflammation, bleeding wounds, burns, etc. | Preparations: Roots and rhizome are smoked for mild sedation and vivid dreams or in a powdered form and sniffed. The fresh roots and rhizomes may be chewed, while the roots are boiled, and a decoction is taken orally. Fresh rhizome is also applied externally as a styptic. Applications: Oral administration, sniffed, and externally applied. |

**Table 1.** *Cont.*

| Scientific Names | Family | Common Names | Plant Part Used | Medicinal Uses | Applications and Preparations |
|---|---|---|---|---|---|
| *Xysmalobium undulatum* (L.) W.T. Aiton | Apocynaceae | White Bush (E), Ishongwe (Z), and Poho-tsehla (S) | Entire plant/roots and leaves | Used as an anti-diarrhoeal, spasmolytic, and wound-healing; used topically to sores and wounds. | Preparations: The powdered root is used to cure wounds and abscesses, as an anti-diarrhea remedy, and as a snuff. The dry, powdered root and its extract possess an antispasmodic effect and are said to be an effective cure for unpleasant menstrual cramps. In youngsters, it is used as a vermifuge, as well as a decongestant and for headaches. Both the roots and the leaves are cooked, and the decoction is consumed. Application: Taken via oral administration. |
| *Asparagus africanus* Lam. | Asparagaceae | uMathunga (X) | Roots | Used for sores and wounds, flatulence, and colic. Cleanses the blood to assist HIV/AIDs patients. | Preparations: Warm bulb decoctions in water or milk are frequently taken orally for many weeks. Mode of action: "Taken per os". |
| *Aloe aristata* Haw. | Asphodelaceae | Umathithibala (Z) | Leaf | Used in the treatment of infections, internal parasites, and digestive ailments and as a dressing for wounds. | Preparations: Aloe is mixed with water to wash wounds and sores for a refreshing effect. Also used as a laxative. The inner pulp is used externally to relieve skin discomforts. Applications: Applied on skin; Taken as laxative. |
| *Cotyledon orbiculata* L. | Crassulaceae | Pig's Ear (E) and Morianna wa di-tsebe/sereledi (S) | Leaves/Stem | Used for inflammation, removes warts, and treats epilepsy, internal parasites, skin ailments, and various diseases. | Preparations: Corns and warts are treated using fleshy leaves. The leaves' juice is used as a drop for earaches and toothaches, as well as a heated poultice for boils and inflammation. Fresh leaf juice is taken twice or three times a day in half-cup amounts. The leaves are used to cure boils, earaches, and inflammation, as a vermifuge, and as a heated poultice. To cure acne and other skin disorders, roots and rhizomes are cooked in water and applied externally. Applications: Applied externally, inhaled, and via oral administration. |
| *Dioscorea elephantipes* (L'Her.) Engl. | Dioscoreaceae | Elephant's foot (E), and Ingweva (Z) | Whole plant | Used for sores, wounds, syphilis, and peptic ulcers. | Preparations: Before boiling, the whole plant is submerged in water for three days. The skin is massaged with peeled or grated root. It has the potential to be utilized as a contraceptive. Applications: Taken orally and applied externally or topically as an ointment. |

**Table 1.** *Cont.*

| Scientific Names | Family | Common Names | Plant Part Used | Medicinal Uses | Applications and Preparations |
|---|---|---|---|---|---|
| *Dioscorea sylvatica* Eckl. | Dioscoreaceae | Wild Yam (E) Ingefu/Uskolpati (Z), Usikolipati (X), and Skilpadknol (A) | Whole plant | Used for skin problems. | Preparations: Human and animal sores and wounds are treated with water heated in the scooped-out tuber. The fresh, peeled rhizome is rubbed on the skin. Infusions are takenduring pregnancy to ensure health, as oral contraceptives, and to treat nervous spasms. Applications: Applied externally on skin; taken via oral administration. |
| *Elephantorrhiza elephantina* (Burch.) Skeels | Fabaceae | Elephant's foot/Mosquito plant/(E), Mositsane (S), and Intolwane (Z) | Underground parts | Used for sunburn, acne, burns, and rash. | Preparations: The grated root is steeped in water for 24 h or more then strained and ready for external use. For internal use, it must be boiled for 10 min, and small quantities must be taken 3 times a day. When treating acne, a warm infusion is used to hold the face in a vapor. Sunburn may be treated using the subterranean sections, and acne can be treated with root infusions. Applications: Taken orally; applied topically as lotion or ointment. |
| *Afzelia quanzensis* Welw. | Fabaceae | Lucky bean (E), umDlavusa (Z), umHlavusi (X), and Peulmahonie (A) | Bark/root | Used for burns and warts. | Preparations: Cold water infusions of powdered bark are taken orally. Applications: Oral administration. |
| *Eucomis bicolar* Baker. | Hyacinthaceae | uMbola (Z) | Bulbs | Used for skin ailments, burns, and wounds. | Preparations: Decoctions and infusions are prepared. Applications: Oral administration. |
| *Eucomis autumnalis* (Mill.) Speta. Chitt | Hyacinthaceae | Pineapple flower/Pineapple lily (E), and uMakhandakantsele (Z) | Bulbs/roots | Used for ores and wounds. | Preparations: Warm bulb decoctions in water or milk are frequently taken orally for many weeks. Powdered sun-dried roots are steeped in water before being taken orally until the patient is healed. Applications: Taken orally. |
| *Merwilla plumbea* (Lindl.) Speta (Blue squil) | Hyacinthaceae | Wild squill/Blue hyacinth (E), and Inguduza (Z) | Bulbs | Used in the treatment of skin conditions. Bulb decoctions are used as enemas and purgatives and for boils, sprains, and fractures. | Preparations: Externally, ointments made from fresh bulbs are used to treat skin conditions such as boils and ulcers. Powdered ash from burned plants and bulbs is used on wounds and scrapes as well as sprains and fractures. Bulb decoctions/infusions are prepared from bulbs, gently warmed, and given orally till the patient is healed. Applications: Taken as oral administration, applied topically as ointment. |

**Table 1.** *Cont.*

| Scientific Names | Family | Common Names | Plant Part Used | Medicinal Uses | Applications and Preparations |
|---|---|---|---|---|---|
| *Hypericum aethiopicum* Thunb. | Hypericaceae | John's Wort/Two days (E), uNsukumbili (Z), and Bohoho (S) | Leaves/bark | Used for stomach ulcers and complaints, fever, backache and as an antidepressant and diuretic. | Preparations: The grated plant is boiled for 20 min and then strained and must be taken 3 times a day. Alternatively, half a cup of boiling water must be poured over two spoons of the powdered plant and then strained after 10 min. Applications: Taken via oral administration. |
| *Hypoxis hemerocallidea* Fisch. Mey. & Ave & Ave-Lall. | Hypoxidaceae | African potato (E), Inkomfe (Z), Ilabatheka (X). | Corm | Used as an immune stimulant and for mouth and oral ulcers, colds and flu, cancer, and tumors. | Preparations: The corm is diced, boiled, and taken internally. The dosage depends on the discretion of the person with sores. Applications: Taken orally. |
| *Lycopodium clavatum* L. | Lycopodiaceae | Clubmoss/Belly powder (E), and uMnwele (Z) | Whole plant | Used for wounds, sores, warts, and other skin ailments. | Preparations: The whole part is ground into powder, boiled, and taken internally, or the roots applied externally on burns. Applications: Taken orally. |
| *Themeda triandra* forssk. var. burchellii (Hack.) Stapf | Poaceae | Seboku (S) | Leaves | Used for eczema and skin allergies. | Preparations: Infusion of ground leaves and water is drunk for stomach pains. Applications: Taken as oral administration. |
| *Pentanisia prunelloides* (Klotzsch ex Eckl. & Zeyh.) Walp. | Rubiaceae | Wild verbena/broad leaved Pentanisia (E), Icimamlilo (Z), Setimamollo (S), and Sooibrandbossie (A) | Root | Root decoctions are used to treat burns and swellings, both internally and topically. | Preparations: Boils and other easily accessible inflammations are treated with a poultice made from the heated leaf. To soften and remove stubborn corns and warts, the fleshy section of the leaf is used. A dried leaf is used as a protective charm and a toy for orphaned Basotho children. Also used to treat insect stings and bites when combined with *Dicoma anomala*. Applications and Mode of action: Applied topically as ointment; taken orally. |
| *Hermannia depressa* N.E.Br. | Sterculiaceae | Doll's Rose (E) and seletjhane (S) | Root | Used for headaches and wound healing. | Preparations and Application: Externally applied roots for burns, swellings, and irritated wounds. |

Common names: (E)—English; (Z)—IsiZulu; (S)—Sesotho; (A)—Afrikaans; (X)—IsiXhosa.

The families Hyacinthaceae, Fabaceae, and Dioscoreaceae had the most plant species used in the treatment of skin infections in the study area, while *Cotyledon orbiculata* (0.75), *Dioscorea sylvatica* (0.63), and *Lycopodium clavatum* (0.50) had the highest relative frequency of citation values and were chosen for a phytochemical analysis and pharmacological investigation (Table 2).

**Table 2.** List of plants frequently used in the management of skin diseases in the eastern Free State.

| Plant Species | Frequency of Citation | Relative Frequency of Citation |
|---|---|---|
| *Cotyledon orbiculata* | 6 | 0.75 |
| *Dioscorea sylvatica* | 5 | 0.63 |
| *Lycopodium clavatum* | 4 | 0.50 |
| *Hermannia depressa* | 3 | 0.37 |
| *Pentanisia prunelloides* | 3 | 0.37 |
| *Merwilla plumbea* | 2 | 0.25 |
| *Eucomis bicolar* | 2 | 0.25 |
| *Xysmalobium undulatum* | 2 | 0.25 |

*3.2. Phytochemical Analysis*

The results of the phytochemical analysis are presented in Tables 3–5. Tannins, flavonoids, alkaloids, anthraquinones, steroids, cardiac glycosides, and terpenoids were confirmed in *C. orbiculata*, *L. clavatum*, and *D. sylvatica* extracts (Table 1). The *C. orbiculata* stem extract displayed a phenolic content of 1.20 ± 0.64 mg GAE/g, while *D. sylvatica* acetone and aqueous extracts exhibited good total phenolic contents of 1.48 ± 0.13 and 1.00 ± 0.13 mg GAE/g, respectively. Additionally, an *L. clavatum* ethanol extract displayed a higher phenolic content (1.50 ± 0.13 mg GAE/g) than its acetone extract with a phenol content of 0.92 ± 0.13 mg GAE/g. Table 4 further presents the chemical compositions of the plants from previous studies.

**Table 3.** Phytochemical analysis of plants used against skin ailments in the eastern Free State.

| Plant Name | Extract | Tannins | Terpenoids | Saponins | Flavonoids | Cardiac Glycosides | Alkaloids | Anthroquinones | Steroids |
|---|---|---|---|---|---|---|---|---|---|
| *Cotyledon orbiculata* (Stem) | MeOH | + | − | − | + | + | − | − | − |
| | EtOH | + | − | − | + | + | − | − | − |
| | Acetone | + | + | − | + | − | + | − | − |
| | $H_2O$ | + | − | + | + | − | − | − | − |
| *C. orbiculata* (Leaves) | MeOH | + | − | − | − | − | + | + | − |
| | EtOH | − | − | − | − | − | + | + | − |
| | Acetone | − | + | − | + | − | + | + | − |
| | $H_2O$ | − | + | − | − | − | − | − | − |
| *Dioscorea sylvatica* | MeOH | − | + | + | − | − | + | − | − |
| | EtOH | − | + | − | − | − | + | + | − |
| | Acetone | − | + | − | + | − | − | + | − |
| | $H_2O$ | − | − | − | − | − | − | + | + |
| *Lycopodium clavatum* | MeOH | + | − | − | + | + | − | − | − |
| | EtOH | + | − | − | + | − | + | − | + |
| | Acetone | + | + | − | + | − | − | − | − |
| | $H_2O$ | + | + | + | + | − | − | − | − |

+ present; − absent.

**Table 4.** Chemical compositions of the plants used against skin ailments in the eastern Free State.

| Plant Species | Family | Common Name | Plant Part Used | Chemical Composition |
|---|---|---|---|---|
| *Cotyledon orbiculata* L. | Crassulaceae | Pig's Ear (E); Morianna wa di-tsebe/Sereledi (S) | Leaves/stems | Chemical characterisation—a total of 32 compounds were found in an early somatic embryo, 33 compounds in a mature somatic embryo, 32 compounds in a germinated somatic embryo extract, via UHPLC/MS/MS ex-vitro studies [21–23]. A flavone derivative, linoleamide, and oleamide were detected from extracts and fatty amides [23]. |
| *Dioscorea sylvatica* Eckl. | Dioscoreaceae | Wild Yam (E); Ingefu/Uskolpati (Z); Skilpadkol (A) | Whole plant | Previous phytochemical investigations of the genus via TLC and HPLC revealed the following: alkaloid dioscorin sapogenins, sterodial saponins, and related, steroids, glycosides of diosgenin, dioscin, alkaloids (dioscorine and dihydrodioscorine) [24,25]. Other compounds found were oxalate salts and calcium oxalate raphide (determined via a light microscopic observation in the tuber/bulb) [24,25]. |
| *Lycopodium clavatum* | Lycopodiaceae | Clubmoss/Belly powder (E); uMnwele (Z) | Whole Plant | Methylation and demethylation were determnined (in vivo) via UPLC-Q TOF/MS [26]. Lycojaponicumin C, huperzine E, and lycopodine (in-vitro) were determined via MS and MS/MS [26]. |

Common names: (E)—English; (Z)—IsiZulu; (S)—Sesotho; (A)—Afrikaans; UHPLC/MS/MS—ultra high-performance liquid chromatography–tandem mass spectrometry method; TLC—thin layer chromatography; HPLC—high-performance liquid chromatography; UPLC-Q TOF/MS—ultra high performance liquid chromatography with quadrupole time-of-flight mass spectrometry; MS—MS/MS—tandem mass spectrometry.

**Table 5.** Total phenolic content of plants used against skin ailments in the eastern Free State.

| Plant Name | Extracts (Values in mg GAE/g) Percentage (%*w/w*) | | | |
|---|---|---|---|---|
| | **Methanol** | **Ethanol** | **Acetone** | **Aqueous** |
| *C. orbiculata* (stem) | $1.48 \pm 0.64$ | $1.20 \pm 0.64$ | $1.48 \pm 0.64$ | $0.78 \pm 0.64$ |
| *D. sylvatica* | $0.56 \pm 0.13$ | $0.75 \pm 0.13$ | $1.48 \pm 0.13$ | $1.00 \pm 0.13$ |
| *L. clavatum* | $0.75 \pm 0.13$ | $1.50 \pm 0.13$ | $0.92 \pm 0.13$ | $0.67 \pm 0.13$ |
| Gallic acid (standard reference) | $0.87 \pm 0.07$ | | | |

Total phenolics as the gallic acid equivalent GAE mg/g of extracts. Results are given as mean $\pm$ S.E (standard error) values. *p*-values less than $p < 0.05$ are considered significant.

### 3.3. Antibacterial Activity

Table 6 shows the MIC values of *C. orbiculata*, *D. sylvatica*, and *L. clavatum* extracts. Against the test microorganisms, all of the tested plant extracts had varying degrees of antibacterial activity. The best activity was displayed by the *L. clavatum* acetone extract against *S. aureus* and *P. aeruginosa* at 0.39 and 0.098 mg/mL, respectively, when compared with the reference drug (neomycin). The *L. clavatum* aqueous extract also displayed good

antibacterial activity against *E. coli* at 0.78 mg/mL. Appreciable antibacterial activities were also observed with the acetone and methanol extracts prepared from *C. orbiculata* leaves against *P. aeruginosa* at a concentration of 0.78 mg/mL. Moderate activity was observed with the *C. orbiculata* acetone stem extract against *P. aeruginosa* (1.56 mg/mL).

**Table 6.** Antibacterial activity of plants used against skin ailments in the eastern Free State (MIC values in mg/mL).

| Plant Name | Plant Part Used | Extract Yield (g) | Extract | Bacterial Strains | | | | |
|---|---|---|---|---|---|---|---|---|
| | | | | *K.p.* | *B.p.* | *E.c.* | *S.a.* | *P.a.* |
| *C. orbiculata* | Stem | 143.96 | Ace | 6.25 | 6.25 | 6.25 | 6.25 | 1.56 |
| | | 139.65 | EtOH | 5.21 | 6.25 | 6.25 | 2.15 | 6.25 |
| | | 135.42 | MeOH | 6.25 | 6.25 | 6.25 | 6.25 | 3.125 |
| | | 148.95 | dH$_2$O | 6.25 | 6.25 | 6.25 | 6.25 | 6.25 |
| *C. orbiculata* | Leaf | 176.62 | Ace | 2.08 | 1.56 | 1.56 | 3.125 | 0.78 |
| | | 174.34 | EtOH | 6.25 | 3.125 | 1.56 | 6.25 | 6.25 |
| | | 163.24 | MeOH | 2.60 | 3.125 | 3.125 | 3.125 | 0.78 |
| | | 181.32 | dH$_2$O | 4.70 | 4.70 | 1.56 | 1.56 | 1.56 |
| *D. sylvatica* | Whole plant | 173.24 | Ace | 3.125 | 6.25 | 12.5 | 6.25 | 3.125 |
| | | 184.36 | EtOH | 6.25 | 6.25 | 6.25 | 6.25 | 3.125 |
| | | 163.24 | MeOH | 3.125 | 3.125 | 3.125 | 3.125 | 3.125 |
| | | 167.64 | dH$_2$O | 12.5 | 12.5 | 12.5 | 12.5 | 12.5 |
| *L. clavatum* | Whole plant | 153.32 | Ace | 6.25 | 3.125 | 12.5 | 0.39 | 0.098 |
| | | 143.62 | EtOH | 3.125 | 1.56 | 1.56 | 1.56 | 6.25 |
| | | 148.32 | MeOH | 12.5 | 12.5 | 1.56 | 1.56 | 6.25 |
| | | 155.24 | dH$_2$O | 12.5 | 6.25 | 0.78 | 3.125 | 12.5 |
| | Neomycin µg/mL (control) | | | 0.098 | 0.098 | 0.098 | 0.098 | 0.098 |

*K.p.*, *Klebsiella pneumoniae*; *B.p.*, *Bacillus pumilus*; *E.c.*, *Escherichia coli*; *S.a.*, *Staphylococcus aureus*; *P.a.*, *Pseudomonas aeruginosa*. Ace, acetone; EtOH, ethanol; MeOH, methanol; dH$_2$0, distilled water.

### 3.4. Antifungal Activity

Table 7 presents antifungal activity results for the *C. orbiculata*, *D. sylvatica* and *L. clavatum* extracts. Extracts prepared from *C. orbiculata* and *D. sylvatica* displayed the best antifungal activity against both *C. albicans* and *T. mucoides*, with MIC values ranging between 0.39 and 1.56 mg/mL. The ethanolic extract from *L. clavatum* also showed good activity at 1.56 mg/mL.

### 3.5. Antioxidant Activity

Table 8 and Figure 1 show the antioxidant properties of the screened extracts. Plant species which showed a higher and stronger antioxidant activity than that of ascorbic acid were the *C. orbiculata* methanol extract (0.10 ± 0.03 µg/mL) and *D. sylvatica* aqueous extract (0.12 ± 0.03 µg/mL). Good antioxidant activity was also observed with the *L. clavatum* aqueous extract with an IC$_{50}$ value of 0.18 ± 0.02 µg/mL. The *L. clavatum* ethanol and methanol extracts displayed antioxidant activity at IC$_{50}$ values of 0.24 ± 0.04 µg/mL and 0.25 ± 0.06 µg/mL, respectively.

The plant extracts' total antioxidant capacity was measured. The following was the rank of total antioxidant capacity: *L. clavatum* > *C. orbiculata* > *D. sylvatica*. The total capacity of antioxidant activity was found to be the best in the *D. sylvatica* water extract, which had an IC$_{50}$ value of 0.03 ± 0.09 µg/mL, followed by ethanol (0.04 ± 0.03 µg/mL) and methanol (0.08 ± 0.0 L µg/mL) extracts. The aqueous extract of *L. clavatum* also displayed a total antioxidant capacity of 0.03 ± 0.09 µg/mL.

**Table 7.** Antifungal activity of plants used against skin ailments in the eastern Free State (MIC values in mg/mL).

| Plant Name | Part Used | Extract Yield (mg) | Extract | Fungal Strains | |
|---|---|---|---|---|---|
| | | | | *C. albicans* | *T. mucoides* |
| *C. orbiculata* | Stem | 146.98 | Ace | 0.78 | 1.56 |
| | | 174.56 | EtOH | 0.78 | 1.56 |
| | | 170.09 | MeOH | 0.78 | 1.56 |
| | | 186.98 | dH$_2$O | 6.25 | 1.56 |
| *D. sylvatica* | Whole plant | 168.57 | Ace | 0.39 | 0.39 |
| | | 134.45 | EtOH | 1.56 | 2.08 |
| | | 109.87 | MeOH | 0.39 | 0.39 |
| | | 197.65 | dH$_2$O | 1.56 | 3.125 |
| *L. clavatum* | Whole plant | 175.36 | Ace | 3.125 | 3.125 |
| | | 127.95 | EtOH | 1.56 | 2.60 |
| | | 143.89 | MeOH | 3.125 | 3.125 |
| | | 145.36 | dH$_2$O | 3.125 | 3.125 |
| | Gentamycin μg/mL (control) | | | 0.049 | 0.049 |

Ace, acetone; EtOH, ethanol; MeOH, methanol; dH$_2$0, distilled water.

**Table 8.** Antioxidant activity of plants used against skin ailments in the eastern Free State (IC$_{50}$ values in μg/mL).

| Plant Name | DPPH Assay (IC$_{50}$ Values in μg/mL) | | | | Total Antioxidant Capacity (IC$_{50}$ Values in μg/mL) | | | |
|---|---|---|---|---|---|---|---|---|
| | Methanol | Ethanol | Water | Acetone | Methanol | Ethanol | Water | Acetone |
| *C. orbiculata* | 0.10 ± 0.03 | 0.37 ± 0.188 | 0.44 ± 0.198 | 0.20 ± 0.05 | 0.44 ± 0.157 | 0.58 ± 0.124 | 0.36 ± 0.126 | 0.62 ± 0.267 |
| *L. clavatum* | 0.25 ± 0.06 | 0.24 ± 0.04 | 0.18 ± 0.02 | 0.57 ± 0.33 | 0.11 ± 0.04 | 0.29 ± 0.08 | 0.03 ± 0.09 | 0.08 ± 0.04 |
| *D. sylvatica* | 0.40 ± 0.04 | 0.28 ± 0.01 | 0.12 ± 0.03 | 0.28 ± 0.02 | 0.08 ± 0.01 | 0.04 ± 0.03 | 0.03 ± 0.09 | 0.14 ± 0.061 |
| Ascorbic acid | 0.27 ± 0.11 | | | | Gallic acid: 0.135 ± 0.365 | | | |

IC$_{50}$ values for methanol, ethanol, aqueous, and acetone extracts. IC$_{50}$: mean of duplicate assays, mean ± SEM (standard error of mean). $p < 0.05$ was considered significantly different.

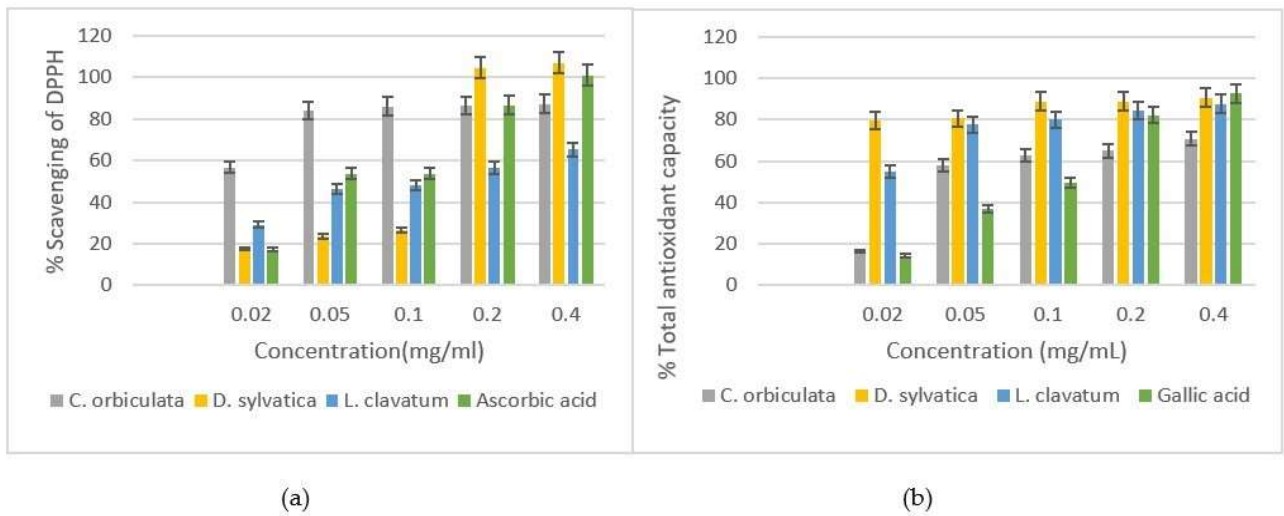

**(a)**　　　　**(b)**

**Figure 1.** (**a**) DPPH radical scavenging activity of methanol extracts; (**b**) total capacity of antioxidant activity of methanol extracts.

*3.6. Anti-Inflammatory Activity*

All of the extracts examined inhibited the 5-LOX enzyme more effectively than the conventional anti-inflammatory medication (NDGA). NDGA's IC$_{50}$ value in this investigation was 0.58 ± 0.35 μg/mL. The IC$_{50}$ value for the *C. orbiculata* ethanol ex-

tract was $0.09 \pm 0.02$ μg/mL, whereas the $IC_{50}$ value for the *C. orbiculata* methanol extract was $0.26 \pm 0.04$ μg/mL. The ethanol extract of *L. clavatum* had an $IC_{50}$ value of $0.02 \pm 0.08$ μg/mL (Table 9 and Figure 2). All of these $IC_{50}$ values were lower than the norm and statistically different ($p < 0.05$).

**Table 9.** Anti-inflammatory activity of plants used against skin infections in the eastern Free State ($IC_{50}$ values in μg/mL).

| Plant Name | 5-Lipoxygenase Assay (μg/mL) | | | |
|---|---|---|---|---|
| | **Methanol** | **Ethanol** | **Water** | **Acetone** |
| *C. orbiculate* | $0.26 \pm 0.04$ | $0.09 \pm 0.02$ | $0.47 \pm 0.24$ | $0.45 \pm 0.64$ |
| *L. clavatum* | $0.16 \pm 0.02$ | $0.02 \pm 0.08$ | $0.26 \pm 0.56$ | $0.04 \pm 0.32$ |
| *D. sylvatica* | $0.49 \pm 0.364$ | $0.25 \pm 0.54$ | $0.25 \pm 0.54$ | $0.24 \pm 0.44$ |
| NDGA (control) | $0.58 \pm 0.35$ | | | |

$IC_{50}$ values for methanol, ethanol, aqueous, and acetone extracts. $IC_{50}$: mean of duplicate assays, mean $\pm$ SEM (standard error of mean). $p < 0.05$ was considered significantly different.

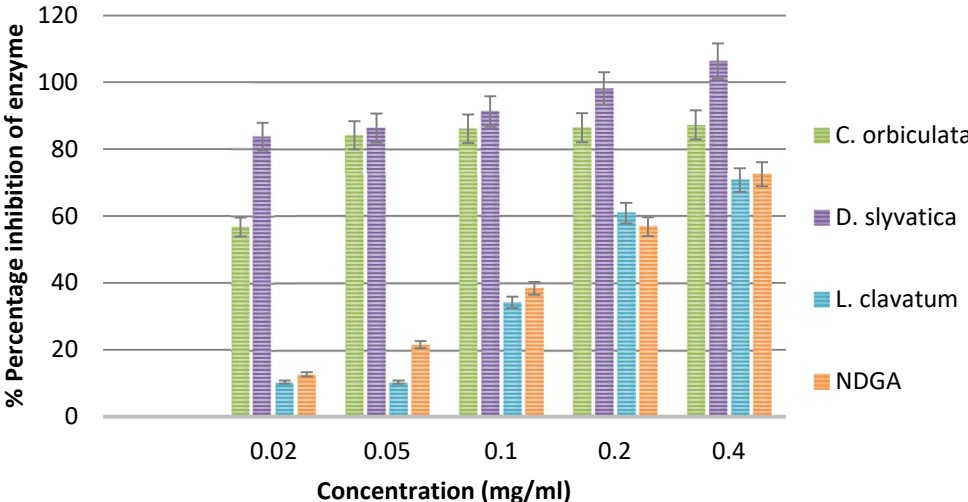

**Figure 2.** The percentage inhibition of 5-lipoxygenase by the methanol plant extracts in relation to NDGA.

## 4. Discussion

### 4.1. Ethnobotanical Survey

To minimize knowledge degradation and improve recording and digitization, the use of medicinal plants in the treatment of a variety of ailments has been recorded in many regions of the globe [27]. It was discovered in this study that some plants were known by many common names. This is due to the fact that various cultures call the same plant by different names. Traditional healers/herbalists also supplied knowledge about the herbal treatments' uses, preparation methods, and methods of application. The bark, leaves, roots, and bulbs or tubers are usually gathered to make medication. This corresponds to the finding of Louw et al. [28]. In this study, the most frequently utilized plant components were reported to be the bark, stem, roots, rhizomes, corms, and bulbs. Leaves and fruits were the least used plant parts in this research, which contrasts with numerous ethnobotanical surveys in which leaves were the most commonly used portions in treating various health issues [29]. The survey revealed that decoction or concoction preparations were consumed orally, with topical application of extracts on wounds and sores employed concurrently. The most common techniques of preparing herbal treatments are decoctions and infusions [30]. Infusion was found to be the most efficient method of preparation since it requires just a few ingredients: simply heated water and the plant

material [31]. Cleaning wounds with aqueous extracts produced from plants and treating wounded skin/wounds with poultices were among the procedures used.

### 4.2. Phytochemical Analysis

The *C. orbiculata*, *L. clavatum*, and *D. sylvatica* extracts were individually tested and shown to contain different phytochemicals which included tannins, saponins, triterpene steroid, reducing sugar, and cardiac glycosides. The presence of tannins suggests the usefulness of a plant extract as an anti-diarrheic and antihemorrhagic agent [32]. Extracts obtained using organic solvents for extraction were observed to show the highest total phenolic contents. This may be due to the fact that organic solvents have a higher polarity index than other solvents and are able to extract more phenolic and polyphenolic compounds [33,34]. The leaves of *C. orbiculata* were reported to contain tannins, saponins, triterpene steroid, reducing sugar and cardiac glycosides in a study undertaken by Orhan [35]. The genus of *Lycopodium* is known to be rich in alkaloids with high toxicity; this may also contribute to the antimicrobial activity of the *L. clavatum* extracts. It is also reported that the *Lycopodium* genus also contains various phenolic acids such as dihydrocaffeic, vanilla syringic, and phenolic acids, which are known to possess antimicrobial activity against a variety of microorganisms [36].

### 4.3. Antibacterial Activity

All screened plant extracts displayed some level of antibacterial activity against the test microorganisms. The best activity was displayed by the *L. clavatum* acetone extract against *S. aureus* and *P. aeruginosa* at 0.39 and 0.098 mg/mL, respectively. A threshold for a good antibacterial activity was noticeable or recorded at 0.098 mg/mL in relation to that of neomycin. The aqueous extract also displayed good antibacterial activity against *E. coli* at 0.78 mg/mL. The good antibacterial activities recorded may be attributed to the phytochemical constituents that were detected in the extracts. The genus *Lycopodium* contains various phenolic/carboxylic acids which are known to possess antimicrobial activity against a variety of microorganisms [35].

### 4.4. Antifungal Activity

The antifungal properties of various extracts examined revealed that *C. orbiculata* and *D. sylvatica* had significant activity against both *C. albicans* and *T. mucoides*, with MIC values ranging from 0.39 to 1.56 mg/mL. This is similar to the findings of Orhan et al. [35], who documented potent antifungal activities of these plant extracts against *C. albicans* and *T. mucoides*. The antifungal activities of medicinal plant extracts have been adduced to the presence of secondary metabolites like saponins, flavonoid, phenols, and alkaloids in those extracts [30]. The antifungal activity recorded for these plant extracts justifies their traditional use in the management of skin infections in the eastern Free State. A study by Kelmanson et al. [37] on *D. sylvatica* tuber bark showed higher antibacterial and antifungal activities. It seemed logical that the tuber bark should possess the highest antifungal and antibacterial activity as this is the part of the plant that is in constant contact with soil.

### 4.5. In Vitro Antioxidant Activity

All the plant extracts analyzed in the DPPH assay possessed antioxidant activity. A lower or reduced $IC_{50}$ value suggested better antioxidant activity. Because ascorbic acid is a well-known antioxidant that may scavenge DPPH radicals [19], it was employed as a baseline. Antioxidant activity may be due to the presence of phenolic content and other phytochemicals in the plant extracts [38]. Orhan et al. [35] also reported on the antioxidant activity of *L. clavatum* extracts. The Dioscoreaceae family is known for its strong antioxidant activity and high phenolic content [39,40]. There exists a positive relationship between antioxidant potential and the phenol antioxidant index [41]. The total antioxidant capacity (TAC) recorded in this study showed that many of the tested samples had a better TAC compared to the standard. These results suggest that these plant extracts possess the

capacity to scavenge reactive oxygen species and free radicals that delay wound healing or are involved in the etiopathogenesis of skin infections.

### 4.6. In Vitro Anti-Inflammatory Activity

All the plant extracts tested had appreciable inhibitory activities against the 5-LOX enzyme when compared to a standard drug. Because of its reported high inhibitory effects on this enzyme, NDGA was utilized as a reference drug in investigations of 5-LOX inhibition [42,43]. A higher inhibitory activity was associated with a lower or reduced $IC_{50}$ value [44,45]. Amabeoku and Kabatende [46] had previously reported the in vivo anti-inflammatory activity of *C. orbiculata* using a rat model. The potential in vitro anti-inflammatory activities recorded for the tested medicinal plants in this study may suggest that these medicinal plants may be useful in the management of skin diseases.

## 5. Materials and Methods

### 5.1. Study Area

This study was conducted in the eastern part of the Free State Province (Figure 3). The eastern Free State in South Africa is an interior bean-shaped province that covers around 10.6% of the country's total territory. Agriculture, mining, and industry are the mainstays of the economy.

Approximately 90% of the province is cultivated for agricultural production. The province produces approximately 34% of South Africa's total maize, 37% of its wheat, 53% of its sorghum, 33% of its potatoes, 18% of its red meat, 30% of its groundnuts, and 15% of its wool [47].

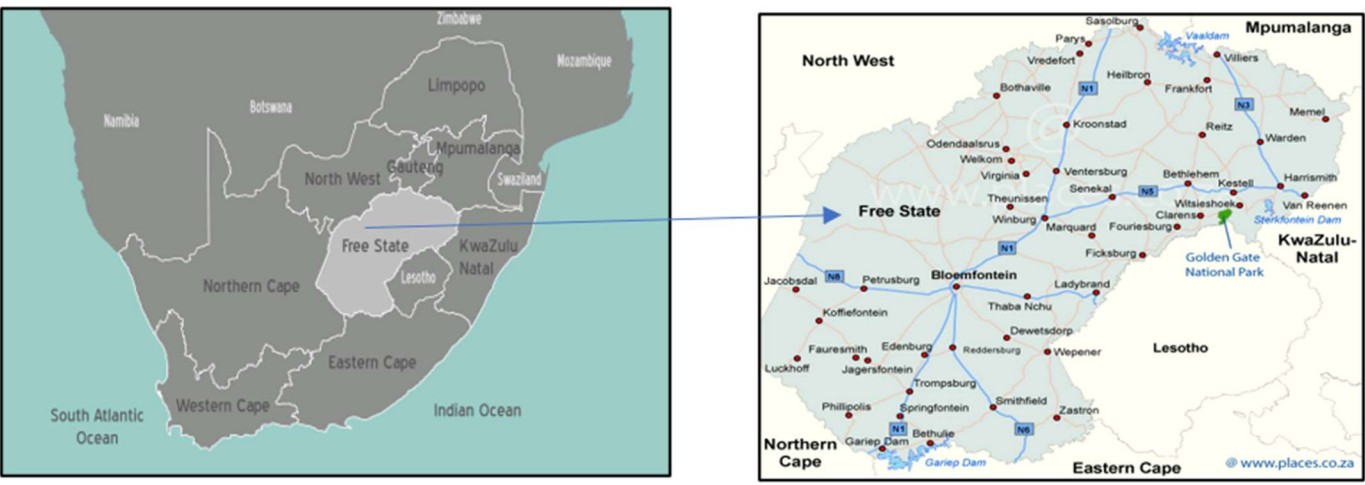

**Figure 3.** Map showing the Free State Province. Arrow indicates the study area. Source: SA [48].

The grasslands with semi-arid vegetation in the south and the hilly regions in the east comprise the majority of the Free State province, with the eastern part covered by the Eastern Free State Sandy Grassland, Basotho Montane Shrubland, and Northern Drakensberg Highland Basalt Grass biomes [47]. The study area has two vegetation types, the Eastern Free State Sandy Grassland and the Basotho Montane Shrubland. The area reaches an altitude of 1520–1800 m, even reaching altitudes of 2020 m in some places within the region. It consists of cool to very cold winters and warm to hot summer days, with snowfall almost every year. The area is a summer rainfall region with an annual rainfall of more than 720 mm along the Maloti Mountains range and can receive rainfall of more than 1400 mm, particularly in wet years [47]. The area is characterized by a high number of unemployed rural dwellers.

### 5.2. Ethnobotanical Survey

An ethnobotanical survey was conducted from March 2014 to August 2014. The interviews were conducted using a structured questionnaire; they were discussed on an individual basis and explained by an interpreter conveying messages in Sesotho or IsiZulu. This was due to a misinterpretation of the English language. A total of ten informants included two traditional healers, four herbalists, and four vendors, comprising eight females and two males. The interviews were conducted at herbal markets in the Free State Province, and other commercial specimens were purchased at Durban *Muthi* market.

### 5.3. Plant Collection and Identification

Plants were collected in different areas (Mangaung, Tseseng, Tseki, and Bluegumbosch) with the assistance of the herbalists and traditional healers. The traditional healers/herbalists provided the local names of the plants used (*Cotyledon orbiculata*, *Lycopodium clavatum*, and *Dioscorea sylvatica*) (Table 1); the identification of scientific names was carried out at the University of the Free State, Qwaqwa Campus, by Dr Erwin J.J. Sieben. Voucher specimens were made and stored at the UFS Qwaqwa Campus Herbarium with the ethical approval number UFS-HSD2015/0021.

### 5.4. Intellectual Property Agreement Statement

During the ethnobotanical survey, all traditional healers and herbalists who provided information for this research study were monetarily compensated. Furthermore, informed consent was provided, and an agreement was reached that the information would not be for commercial purposes; it would be recorded in order to educate the people of the Free State Province and the rest of the world about the plants that are used to cure and manage skin problems.

### 5.5. Relative Frequency of Citation (RFC)

The relative frequency of citation was used to assess the local value of each plant species [49]. The RFC was determined using the following formula: The total number of informants divided by 132, the number of informants who indicated the usage of the species (Fc) (N). The RFC was calculated as RCF = Fc/N (Table 2).

### 5.6. Preparation of Plant Extracts

The plant components were chopped into tiny pieces and dried in an oven for 24 h at 40 °C. Using a blender, the dried plant material was blended into a powder. Approximately 40 g of powered material from each of the plants was extracted using 400 mL acetone, ethanol, methanol, and distilled water. This was achieved by shaking the powdered plant material in the solvents for 24 h and filtering the solution through Whatman No. 1 filter paper discs. The filtrates were dried using a rotary evaporator (Cole Parmer SB 1100, Shangai, China).

### 5.7. Phytochemical Analysis

The phytochemicals of *C. orbiculata*, *L. clavatum*, and *D. sylvatica* were determined in the aqueous and powdered plant material by adopting the standard methods described in [50–52]. Plants were individually tested for the presence of alkaloids, flavonoids, terpenoids, saponins, anthraquinones, cardiac glycosides, and tannins. The presence of phytochemicals was determined by a visual observation of color change or the production of a precipitate upon the addition of the prescribed reagent(s).

### 5.8. Determination of Total Phenol Content

The total phenolic content was determined following the method of Singleton [53], with alterations described in [54]. Briefly, a mixture containing 50 μL of plant extract, 50 μL of 10% Folin C reagent, and 150 μL of 7.5% sodium carbonate was prepared and transferred to Eppendorf tubes. For color development, the tubes were vortexed for 15 s

and left at 40 °C for 30 min. The mixture was then incubated for 40 min at 45 °C. After incubating, a spectrophotometer(DR2800, Hach, USA) was used to measure absorbance at 765 nm. As a control, gallic acid was utilized. Extrapolated from the gallic acid curve ($R^2 = 0.9236$), the data were represented as gallic acid equivalents (GAE, mg/g). The tests were conducted twice.

### 5.9. Antibacterial Activity

*Escherichia coli* (ATCC 8739), *Pseudomonas aeruginosa* (ATCC 1958), *Staphylococcus aureus* (ATTC 6538), *Klebsiella pneumoniae* (ATTC 13047), and *Bacillus pumilus* (ATCC 14884) were obtained from [anonymized] and maintained on Mueller–Hinton (MH) agar. The bacterial agents were chosen as the most common wound and skin causal infections.

The minimum inhibitory concentration (MIC) values for the plant extracts with antibacterial activity were determined using Eloff's microplate technique [55]. Plant extract residues were dissolved in extracting solvents at a concentration of 50 mg/mL. All extracts were evaluated at 12.5 mg/mL in 96-well microplates and serially diluted twofold to 0.098 mg/mL before being introduced to each well containing 100 μL of bacterial culture. As a positive control, each bacterial colony was administered 0.1 mg/mL of the antibiotic neomycin. Negative controls included the solvent used to dissolve the plant extracts and bacteria-free wells. All extracts were tested three times. The microplates were covered and incubated for 24 h at 37 °C. Then, 40 μL of p-iodonitrotetrazolium violet (INT) dissolved in water at 0.2 mg/mL was added to the wells as a bacterial growth indicator and incubated at 37 °C for 30 min. The lowest concentration of plant extracts that totally inhibited bacterial growth was recorded as the MIC value (a clear well).

### 5.10. Antifungal Activity

Two fungal strains, *T. mucoides* and *C. albicans*, were used in the test for antifungal activity. These fungal strains were chosen due to their pathogenicity and increasing rates of resistance against antifungal agents. The broth microdilution test, with modifications, was performed [56]. To 400 μL of 24 h old fungal cultures, four milliliters of sterile saline was added. The absorbance, 179, was measured at 530 nm and matched to a 0.5 McFarland standard solution using sterile saline. A 1:1000 dilution with broth (10 μL stock fungal culture: 10 mL broth) was prepared from the prepared stock cultures. The residues from the water extract were dissolved in water, whereas the residues from the organic solvent extract were dissolved in dimethyl sulfoxide (DMSO). At a concentration of 100 mg/mL, all extracts were dissolved. Organic solvent extracts were tested at 6.25 mg/mL and diluted twofold to 0.049 mg/mL, while water extracts were examined at 25 mg/mL and serially diluted to 0.195 mg/mL. In each well, 100 μL of fungal culture was introduced. For each extract, three duplicates were made. The positive control for this experiment was amphotericin B (1.5 mg/mL), whereas the negative controls were wells containing only broth, the solvent used to dissolve the extracts, and a fungal strain with no extract. At 37 °C, the microplates were incubated overnight, and 40 L of 0.2 mg/mL INT solution was added to the wells as a fungal growth indicator and incubated at 37 °C for 30 min.

### 5.11. In Vitro Antioxidant Assay

The 1-1-diphenyl-2-picrylhydrazyl (DPPH) radical scavenging and total antioxidant capacity were used to measure the antioxidant activity of various plant extracts. DPPH is a frequently used technique for assessing the radical scavenging activity of antioxidants [19]. It is claimed to be a stable free radical and is used to measure the radical scavenging activity of antioxidants.

### 5.12. DPPH Radicals Scavenging Assay

The method described by Ursini et al. [57] and Liyana-Pathirana and Shahidi [58] was utilized as a guideline. A volume of 150 mL of 0.4 mM DPPH solution in methanol was gently mixed with 150 mL of plant extract. To dissolve the extract, the reaction mixture was

vortexed completely and kept at room temperature for 30 min. The mixture's absorbance was measured at 517 nm. As a control, ascorbic acid was employed. GraphPad was used to calculate the $IC_{50}$ values. Higher free radical scavenging activity was demonstrated by lower absorbance values in the reaction mixture. All of the tests were conducted three times. The proportion of DPPH radical inhibition was estimated using the following formula [40]:

$$\text{DPPH radical scavenging activity (\%)} = \frac{\text{Abscontrol} - \text{Abssample}}{\text{Abscontrol}} \times 100$$

where Abscontrol is the absorbance of DPPH radical + methanol; and Abssample is the absorbance of DPPH radical + sample extract/standard.

### 5.13. Phosphomolybdenum Total Antioxidant Capacity Assay

The phosphomolybdenum technique was carried out [59]. In 20 mL of distilled water, one milliliter each of 0.6 mM sulfuric acid, 28 mM sodium phosphate, and 4 mM ammonium molybdate was added, and the volume was increased to 50 mL by adding distilled water. Each Eppendorf tube containing 1 mL of molybdate reagent solution and 3 mL of water received 100 microliters of plant extract. These tubes were incubated for 90 min at 95 °C. The absorbance was measured at 765 nm against a reagent blank after normalization at room temperature. As a baseline, ascorbic acid was utilized. The following formula was used to calculate total antioxidant capacity:

$$\text{Total antioxidant capacity (\%)} = \frac{\text{Abscontrol} - \text{Abssample}}{\text{Abscontrol}} \times 100$$

### 5.14. In Vitro Anti-Inflammatory Activity Assay

Each plant extract's anti-inflammatory activity was assessed using the technique described by Baylac and Racine [60] and Trouillas et al. [29]. Linoleic acid was used as the substrate for the 5-lipoxygenase (5-LOX) enzyme. The anti-arachidonate 5-LOX, an antibody produced in rabbits, was used. A standardization process was carried out with the help of the reference sample which comprised a mixture containing 10 μL of plant extract dissolved in a solution of Tween 20 in DMSO, 2.95 mL of phosphate buffer (pH 6.3) pre-warmed in a water bath at 25 °C, and 50 μL of linoleic acid (final concentration, 100 μM). The process was initiated by adding 12 μL of enzyme and 12 μL of phosphate buffer. For 10 min, the absorbance was measured at 234 nm. The reaction rates were evaluated by adding decreasing quantities of extracts. As a positive control, nordihydroguaiaretic acid (NDGA) was employed. GraphPad was used to calculate the $IC_{50}$ value (the concentration at which 50% of the enzyme is inhibited) for each sample test. By comparing the percentage inhibition of enzyme activity to negative controls (Tween® 20/DMSO combination), the percentage inhibition of enzyme activity was estimated. The experiment was carried out twice.

## 6. Conclusions

The present study documents plants used in the treatment of skin diseases in the studied area. Extracts prepared from the most frequently used plants, *C. orbiculata*, *D. sylvatica*, and *L. clavatum*, possessed biologically active phytochemicals such as alkaloids, phenols, terpenoids, saponin, tannin, and flavonoids. The extracts of *C. orbiculata* and *L. clavatum* displayed the best antibacterial activity against *P. aeruginosa*, whereas extracts from *C. orbiculata* and *D. sylvatica* displayed the best antifungal activity against the tested fungal strains. This study demonstrated that *C. orbiculata*, and *L. clavatum* extracts possess antioxidant and anti-inflammatory properties; thus, these plant species may be used to obtain natural products that may be used in the management of skin diseases.

**Author Contributions:** V.M.X. conceptualized and designed the study, collected and analyzed the data, and wrote the manuscript. V.M.X. and L.V.B.-K. conceptualized and designed the study and wrote the manuscript. A.L.A. analyzed data and wrote the manuscript. S.Q.N.L. analyzed data and revised and wrote the manuscript. All authors have read and agreed to the published version of the manuscript.

**Funding:** This research received no external funding.

**Institutional Review Board Statement:** The Ethics Committee granted ethical approval for the ethnob-otanical survey and plant collection with the Ethical Approval number UFS-HSD2015/0021 (approval date: 4 May 2015). Before conducting interviews, all partic-ipants provided their informed consent.

**Informed Consent Statement:** Informed consent was obtained from all subjects involved in the study.

**Data Availability Statement:** All data generated or analyzed during this study are included in this published article.

**Acknowledgments:** The authors are most grateful to the traditional healers and herbalists who provided the information. E. Sieben and A.P. Gold are acknowledged for their assistance with plant identification.

**Conflicts of Interest:** The authors declare that they have no competing interests.

### List of Abbreviations

DMSO—dimethyl sulfoxide; DPPH—1,1-diphenyl-2-picrylhydrazyl; GAE—gallic acid equiva-lents; INT—p-iodonitrotetrazolium violet; 5-LOX—5-lipoxygenase; MIC—minimal inhibitory con-centration; NDGA—Nordihydroguaiaretic acid; OTC—over-the-counter; RFC—relative frequency of citation.

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
