# Peer review of "In Vitro Bioactivities of Plants Used against Skin Diseases in the Eastern Free State, South Africa"

_2037-0164, doi:10.3390/ijpb15010002_

Round 1
Reviewer 1 Report
Comments and Suggestions for Authors
Manuscript No: ijpb-2711055
Comments
The authors of the manuscript “In vitro bioactivities of plants used against skin diseases in the eastern Free State, South Africa” conducted the ethnobotanical survey on plants originated from the Eastern Free State Province of South Africa to provide the date about traditional practice in the treatment of the human ailments.
The comments are:
1. Page 1, Section 1, lines 32-36, the Past Tense should be employed when the results, discussion of the obtained results were commented. Pease, check throughout the text, and make the appropriate changes
2. Table 1, the Column Application and preparations – please, make the presentation of the gathered information uniform, namely as the separately Medicinal uses were given, in this column the date regarding activity should be omitted, while the data regarding of the administration and the type of the used preparations should follow the same pattern, always first mentioned one type of the data – if it is the mode of administration, then in the whole table, first give that information (please, use the appropriate terminology if the activity was expected to be systematic, the mode of administration is “orally” or “per os”, please, change “internally” with the mentioned phrases. After that, the type of preparation might be mentioned. Please, reorganize whole Table 1 according to the comments
3. Table 1, please, give the information what the letters in the parentheses in the column “common names” mean
4. When the plant is the first mentioned the full Latin name should be used (species name, author name and Family name), but afterwards, the short name should be employed. Please, check throughout the text
5. The Legends of the Tables and Figures have to point out that the presented results were from the chosen plants
6. Page 2, line 80, please, check “76”.
7. Page 10, line201, Lycopodium should be in italic
8. Page 12, Subsection 5.3, the plants used in the investigation should be named
9. Page 13, Subsection 5.7 – the brief description of applied methods should be given, besides the literature
10. Page 14, lines 370, 371, why “litres”?
Author Response
|
comments |
Corrections/check-list |
|
Page 1, Section 1, lines 32-36, the Past Tense should be employed when the results, discussion of the obtained results were commented. Pease, check throughout the text, and make the appropriate changes |
Corrected |
|
Table 1, the Column Application and preparations – please, make the presentation of the gathered information uniform, namely as the separately Medicinal uses were given, in this column the date regarding activity should be omitted, while the data regarding of the administration and the type of the used preparations should follow the same pattern, always first mentioned one type of the data – if it is the mode of administration, then in the whole table, first give that information (please, use the appropriate terminology if the activity was expected to be systematic, the mode of administration is “orally” or “per os”, please, change “internally” with the mentioned phrases. After that, the type of preparation might be mentioned. Please, reorganize whole Table 1 according to the comments |
Yes, table corrected. Preparations, is followed by applications. Same pattern followed throughout. Orally taken or oral administration is written throughout. Applied externally- ‘topically’
Page 3- 7. Table 1 |
|
Table 1, please, give the information what the letters in the parentheses in the column “common names” mean |
Yes, corrected.
Common names: (E)- English; (Z)- IsiZulu; (S)-Sesotho; (A)- Afrikaans. |
|
When the plant is the first mentioned the full Latin name should be used (species name, author name and Family name), but afterwards, the short name should be employed. Please, check throughout the text |
Yes, corrected. Species name, followed by Family and then common name. Table 1, page 3-7. |
|
The legends of the Tables and Figures have to point out that the presented results were from the chosen plants |
Table 1 mentioned in parenthesis.
Line 109, page 7. Line 263, page 13 Table 2 mentioned in parenthesis Line 274, page 14 |
|
Page 2, line 80, please, check “76”. |
Corrected, 76 removed. Line 80 page 2 |
|
Page 12, Subsection 5.3, the plants used in the investigation should be named |
Corrected.
Traditional healers/herbalists provided the local name of the plants used (Cotyledon orbiculata, Lycopodium clavatum and Dioscorea sylvatica) Line 261 , page 13. |
|
Page 13, Subsection 5.7 – the brief description of applied methods should be given, besides the literature |
Corrected |
|
Page 14, lines 370, 371, why “litres”? |
Yes, corrected. Line 327. Litres removed, milliliters inserted |
Reviewer 2 Report
Comments and Suggestions for Authors
Dear Editor and authors,
In this review article entitled "In vitro bioactivities of plants used against skin diseases in the eastern Free State, South Africa" the author examines medicinal plants used to treat wounds and skin diseases in the eastern Free State province of South Africa. At first glance, the article is not badly prepared, it is readable and clear, the tables and figures are clearly organised and understandable, the text also looks good, but in my opinion, the article lacks very important analyses that would significantly improve it. Below are my comments:
-The p-value must be in italics throughout the text.
-in situ, in vitro, in vivo, in silico must be italicised, even in the article title!
-All Latin names of microorganisms must also be in italics in the description below the tables - correctly throughout the text.
-Throughout the text - once when you write IC50 and once when you write IC50 with a subscript! Correct throughout the document.
-There is no space below Table 7! It is also not clear to me whether there is a standard deviation or SEM in Table 7 - you will include it in the description below the table!
-Standardise the font in Figures 1 and 2.
-Unify and correct citations and references in accordance with journal requirements!
Overall: The authors of the article should also add the chemical composition of the individual plant extracts, because without such an analysis of the compounds, I do not consider it suitable for publication in such a prestigious journal as IJPB. I consider the addition of the chemical analysis of the extracts to be mandatory!
Comments on the Quality of English Language
Extensive editing of English language is required.
Author Response
|
comments |
Corrections/check-list |
|
The p-value must be in italics throughout the text |
Yes, corrected p-value in italics throughout - p<0.05 |
|
in situ, in vitro, in vivo, in silico must be italicised, even in the article title! |
Yes, corrected in article title and throughout the entire document. |
|
All Latin names of microorganisms must be in italics |
Yes, corrected. C. albicans, T. mucoides, E. coli, K. pneumonia, B. subtilis, P. aeruginosa, S. aureus, K. pneumoniae and B. pumilus. |
|
There is no space below Table 7! It is also not clear to me whether there is a standard deviation or SEM in Table 7 - you will include it in the description below the table! |
Yes, corrected. Page 10 and 11, line 152 and 169. SEM (standard error of means) corrected and stated. |
|
Throughout the text - once when you write IC50 and once when you write IC50 with a subscript! Correct throughout the document. |
Yes, corrected- IC50 corrected, 50 to a subscript. |
|
Standardise the font in Figures 1 and 2. |
Corrected, line 158, page 10 |
|
Unify and correct citations and references in accordance with journal requirements! |
corrected |
|
Overall: The authors of the article should also add the chemical composition of the individual plant extracts, because without such an analysis of the compounds, I do not consider it suitable for publication in such a prestigious journal as IJPB. I consider the addition of the chemical analysis of the extracts to be mandatory! |
Corrected: Qualitative phytochemical analysis was done in this study, but a new table has been inserted with chemical compositions of the plants under study. The organic structure of the compound can be accessed through the referred previous studies and literature. Further assays are still ongoing. |
|
Extensive editing of English language is required |
English language revised |
Round 2
Reviewer 2 Report
Comments and Suggestions for Authors
Dear Editor and authors,
The authors have improved their manuscript considerably.
I only have a comment about the English language, which still needs to be checked.
After improving English, I recommend publish this manuscript it in journal IJMB.
Comments on the Quality of English Language
Minor editing of English language is still required.
Author Response
|
Reviewer 2 comments |
Corrections/check-list |
|
The p-value must be in italics throughout the text |
Yes, corrected p-value in italics throughout - p<0.05 |
|
in situ, in vitro, in vivo, in silico must be italicised, even in the article title! |
Yes, corrected in article title and throughout the entire document. |
|
All Latin names of microorganisms must be in italics |
Yes, corrected. C. albicans, T. mucoides, E. coli, K. pneumonia, B. subtilis, P. aeruginosa, S. aureus, K. pneumoniae and B. pumilus. |
|
There is no space below Table 7! It is also not clear to me whether there is a standard deviation or SEM in Table 7 - you will include it in the description below the table! |
Yes, corrected. Page 10 and 11, line 152 and 169. SEM (standard error of means) corrected and stated. |
|
Throughout the text - once when you write IC50 and once when you write IC50 with a subscript! Correct throughout the document. |
Yes, corrected- IC50 corrected, 50 to a subscript. |
|
Standardise the font in Figures 1 and 2. |
Corrected, line 158, page 10 |
|
Unify and correct citations and references in accordance with journal requirements! |
corrected |
|
Overall: The authors of the article should also add the chemical composition of the individual plant extracts, because without such an analysis of the compounds, I do not consider it suitable for publication in such a prestigious journal as IJPB. I consider the addition of the chemical analysis of the extracts to be mandatory! |
Corrected: Qualitative phytochemical analysis was done in this study, but a new table has been inserted with chemical compositions of the plants under study. The organic structure of the compound can be accessed through the referred previous studies and literature. Further assays are still ongoing. |
|
Extensive editing of English language is required |
English language revised |
